# Effect of a ZrO_2_ Seed Layer on an Hf_0.5_Zr_0.5_O_2_ Ferroelectric Device Fabricated via Plasma Enhanced Atomic Layer Deposition

**DOI:** 10.3390/ma16051959

**Published:** 2023-02-27

**Authors:** Ji-Na Song, Min-Jung Oh, Chang-Bun Yoon

**Affiliations:** Department of Advanced Materials Engineering, Tech University of Korea, Siheung-si 15073, Gyeonggi-do, Republic of Korea

**Keywords:** HfO_2_, ZrO_2_, HZO, seed layer, PEALD, ferroelectric

## Abstract

In this study, a ferroelectric layer was formed on a ferroelectric device via plasma enhanced atomic layer deposition. The device used 50 nm thick TiN as upper and lower electrodes, and an Hf_0.5_Zr_0.5_O_2_ (HZO) ferroelectric material was applied to fabricate a metal–ferroelectric–metal-type capacitor. HZO ferroelectric devices were fabricated in accordance with three principles to improve their ferroelectric properties. First, the HZO nanolaminate thickness of the ferroelectric layers was varied. Second, heat treatment was performed at 450, 550, and 650 °C to investigate the changes in the ferroelectric characteristics as a function of the heat-treatment temperature. Finally, ferroelectric thin films were formed with or without seed layers. Electrical characteristics such as the I–E characteristics, P–E hysteresis, and fatigue endurance were analyzed using a semiconductor parameter analyzer. The crystallinity, component ratio, and thickness of the nanolaminates of the ferroelectric thin film were analyzed via X-ray diffraction, X-ray photoelectron spectroscopy, and transmission electron microscopy. The residual polarization of the (20,20)*3 device heat treated at 550 °C was 23.94 μC/cm^2^, whereas that of the D(20,20)*3 device was 28.18 μC/cm^2^, which improved the characteristics. In addition, in the fatigue endurance test, the wake-up effect was observed in specimens with bottom and dual seed layers, which exhibited excellent durability after 10^8^ cycles.

## 1. Introduction

Dynamic random access memory (DRAM) and flash memory are currently widely used. Since the advent of memory semiconductor devices, DRAM has been extensively used owing to its high speed, high density, long lifespan, and low power consumption. However, DRAM suffers from information volatility, and thus, the need for non-volatile memory has increased. Flash memory has emerged as a non-volatile alternative to DRAM owing to its low noise, small size, and large capacity, in addition to its non-volatility [1,2]. However, the application of flash memory is limited because it requires a long time to write and erase and is difficult to operate at low voltages. Although developments in nanotechnology have resulted in significant advances, the limitations of miniaturization are beginning to impede further developments in the manufacturing of DRAM and flash memory. Next-generation memory technologies are essential to overcome these issues and progress toward the large-scale manufacturing of future memory devices [3].

Typically, next-generation memory devices must exhibit high speed, high density, and non-volatility. Ferroelectric random access memory (FeRAM) satisfies these requirements. FeRAM uses ferroelectricity and is non-volatile in that the recorded data are not erased even when power is cut off. Ferroelectric materials possess permanent polarization even in the absence of an external electric field. In addition, FeRAM exhibits low power consumption and rapid operation. FeRAM has, therefore, emerged as a next-generation memory semiconductor and an effective substitute for DRAM or flash memory [4,5,6].

Representative ferroelectric materials include perovskite materials, such as BaTiO_3_, SrTiO_3_, Pb(Ti,Zr)O_3_ (PZT), and SrBi_2_Ta_2_O_9_ (SBT); however, such materials exhibit ferroelectric properties only at a thickness of ≥100 nm, thus impeding the miniaturization of FeRAM devices. In addition, these materials exhibit low compatibility with the complementary metal–oxide–semiconductor (CMOS) process, further limiting their application.

HfO_2_-based ferroelectric materials have been continuously investigated since ferroelectric properties were first observed in Si-doped HfO_2_ in 2011. Ferroelectric properties were observed in HfO_2_-based ferroelectric materials with a fluorite structure even at a thickness of 10 nm, enabling further miniaturization and densification of HfO_2_-based ferroelectric devices. Ferroelectric properties can also be induced by the addition of various dopants (e.g., Si [7], Zr, Y [8], Al [9], Gd, Sr, and La) to HfO_2_.

Zr-doped HfO_2_ exhibits ferroelectricity over a wide range of compositions. A dopant concentration of ~50% results in the greatest ferroelectricity, suggesting that these doped materials can be manufactured relatively easily. Moreover, ferroelectricity has been observed even at a thickness of <10 nm. Zr-doped HfO_2_ thin films have attracted considerable research interest owing to their large band gap (>5 eV) and compatibility with the CMOS process [10,11,12,13,14,15,16,17,18,19].

In this study, we created a ferroelectric layer with excellent step coverage via plasma enhanced atomic layer deposition (PEALD) [20,21] by leveraging the self-limiting surface reaction to control the thickness of the thin film to within several angstroms. A metal–ferroelectric–metal (MFM)-type capacitor was fabricated using titanium nitride (TiN) as the upper and lower electrodes [22,23], with Hf_0.5_Zr_0.5_O_2_ (HZO) as the ferroelectric material. Three parameters were investigated to determine their impact on the ferroelectric properties of the HZO ferroelectric layer. First, four types of HZO nanolaminates were fabricated [24]; second, heat treatment was applied under three rapid thermal annealing (RTA) conditions; finally, specimens with and without a ZrO_2_ seed layer were fabricated.

The electrical properties of the devices were analyzed and compared, including the I–E characteristics, P–E hysteresis, and fatigue endurance. In addition, the crystallinity and component ratio of the thin films were investigated by analyzing the PEALD-grown ferroelectric layer using X-ray diffraction (XRD), X-ray photoelectron spectroscopy (XPS), and transmission electron microscopy (TEM).

## 2. Materials and Methods

### 2.1. HZO Ferroelectric Device Fabrication

Figure 1 shows the schematics for the production of the MFM device and the structure of the fabricated device. TiN(50 nm)/SiO_2_(100 nm)/Si wafers were used in all experiments. SiO_2_ was formed through dry oxidation, and TiN was deposited by sputtering. The prepared substrate wafers were ultrasonic-cleaned sequentially in acetone, ethanol, and deionized (DI) water for 5 min each.

HZO thin films were grown via PEALD (iOV dx2, iSAC RESEARCH, Daejeon, Republic of Korea) at 250 °C. Tetrakis(ethylmethylamido) hafnium (TEMA-Hf, iChems, Gyeonggi-do, Republic of Korea) and Tetrakis(ethylmethylamido) zirconium (TEMA-Zr, iChems, Gyeonggi-do, Republic of Korea) were used as HfO_2_ and ZrO_2_ precursors, respectively. A plasma voltage of 200 W was applied using O_2_ as the plasma source. The HZO thin film was crystallized by applying heat treatment via RTA (Pyrotech, Gyeonggi-do, Republic of Korea) under an N_2_ atmosphere for 30 s. Finally, a TiN upper electrode with a diameter of 400 µm and a thickness of 50 nm was deposited on the HZO ferroelectric film through radio-frequency sputtering (Sputtering System, BLS, Gyeonggi-do, Republic of Korea). The resistance of the TiN upper electrode deposited by sputtering is approximately 29 ohm/sq.

### 2.2. Variables

A schematic of the ALD deposition process and HZO thin film deposition is shown in Figure 2. The effects of three variables—the HZO nanolaminate thickness, annealing conditions, and presence or absence of a seed layer—were investigated. Each fabrication variable can be found in Figure 3.

To form a thin film according to the first variable, the HZO nanolaminate thickness, the total number of cycles of HfO_2_ and ZrO_2_ was fixed at 120. Super cycles composed of HfO_2_ and ZrO_2_ single layers were set as follows: HfO_2_ 5 cycles–ZrO_2_ 5 cycles (expressed as (5,5)*12), HfO_2_ 10 cycles–ZrO_2_ 10 cycles (expressed as (10,10)*6), HfO_2_ 20 cycles–ZrO_2_ 20 cycles (expressed as (20,20)*3), and HfO_2_ 60 cycles–ZrO_2_ 60 cycles (expressed as (60,60)*1). These super cycles were repeated twelve times, six times, three times, and one time, respectively, and the total number of cycles was fixed at 120 cycles to form four stacked ferroelectric layers [25,26].

The post-deposition annealing method, in which heat treatment is conducted before depositing the upper electrode, was adopted to investigate the effect of the annealing conditions. The heat treatment was performed at 450, 550, or 650 °C for 30 s under an N_2_ atmosphere via RTA (Pyrotech, Gyeonggi-do, Republic of Korea) [27,28].

Finally, four types of specimens were prepared considering the presence or absence of the seed layer to investigate the effect of this layer on the ferroelectric properties: specimens with the ZrO_2_ seed layer formed only on the top (denoted top seed layer), only on the bottom (bottom seed layer), on both the top and the bottom (dual seed layer), and with no seed layer. When the ZrO_2_ seed layer is on the bottom, it is marked as B, when it is only on the top, it is marked as T, and when it is on both the top and bottom, it is marked as D.

### 2.3. Evaluation

The thicknesses of the HZO nanolaminates and ZrO_2_ seed layers were measured by spectroscopic ellipsometry (Elli-SE-U, Ellipso Technology, Gyeonggi-do, Republic of Korea). The crystallinity of the ferroelectric layers according to the heat-treatment conditions and the presence or absence of the seed layer was analyzed by high-resolution X-ray diffractometry (HR-XRD; SmartLab, Tokyo, Japan). The component ratio of the deposited HZO ferroelectric layer was analyzed via XPS (NEXSA, Thermo Fisher Scientific, Seoul, Republic of Korea). The electrical properties of the fabricated MFM capacitor were analyzed using a Keithley 4200A-SCS parameter analyzer. The breakdown voltage of the ferroelectric layer was measured by applying a DC voltage, and the I–E and P–E curves were obtained at a frequency of 1 kHz.

## 3. Results

### 3.1. HZO Thin Film Characteristics

Ellipsometry was used to investigate the thickness of the HZO thin film formed via PEALD. The results of the ellipsometer analysis are shown in Figure 4. The thin film was deposited with a thickness of 1.18 Å per cycle for HfO_2_ and 1.15 Å for ZrO_2_. The thicknesses of the (60,60)*1 and (5,5)*12 thin films were 13.37 nm and 12.64 nm, respectively. Regardless of the existence of the seed layer, the total thickness of the formed thin film tended to decrease with the nanolaminate thickness. Therefore, it was determined that the thickness was slightly reduced owing to a solid solution between the two materials at the interface. Notably, the interface where HfO_2_ and ZrO_2_ are formed is strengthened during the stacking process when the nanolaminate thickness is small [29].

Figure 5 is a cross-sectional TEM image of the D(20,20)*3 specimen annealed at 550 °C. The specimen was processed with a focused ion beam prior to the observation of its cross-section via TEM, which revealed the stacked structure of the device, the thicknesses of the ZrO_2_ seed layer, and the HZO ferroelectric layer deposited through PEALD. The thickness of the HZO ferroelectric layer was ~13 nm, whereas the thickness of the ZrO_2_ seed layers formed on the top and bottom of the ferroelectric layer was ~1 nm. These results are in good agreement with those of the ellipsometry measurements.

The (20,20)*3 HZO thin films heat treated at 550 °C were analyzed via XPS to investigate the component ratio of HZO thin films formed via PEALD. The XPS profiles are shown in Figure 6. The elemental ratio of Hf and Zr was 1:1.25; therefore, the composition of the thin film was determined to be Hf_0.44_Zr_0.56_O_2_.

The nanolaminate and crystallinity of the HZO layer were confirmed using the TEM image in Figure 5, and the crystal phase of each sample was confirmed using HR-XRD analysis. The HR-XRD spectra of the deposited thin film demonstrate the effect of the seed layer and heat-treatment conditions on the crystallinity of the sample (Figure 7). The M phase, with corresponding peaks at 28.5° and 31.7°, exhibits non-ferroelectric properties. In addition, a peak at 30.5° confirms the presence of the O phase, which is known to exhibit ferroelectric properties.

The relative ratio of the O phase to the M phase was calculated using Equation (1) to investigate the formation of the ferroelectric phase of the deposited thin film:
Relative O Phase Ratio = {O(1 1 1)}/{M(−1 1 1) + O(1 1 1) + M(1 1 1)}
(1)


The effect of the seed layer and heat-treatment conditions on the relative ratio of the O phase is shown in Figure 8. The seed layer application presents no evident tendency in its effect on the relative ratio of the O phase; however, the relative ratio of the O phase tended to decrease with the formation of the M phase as the RTA temperature increased, regardless of the presence or absence of the seed layer.

### 3.2. MFM Device Electrical Characteristics

Figure 9 shows the breakdown voltages measured by applying a DC voltage in increments of 0.5 V from 0 V to 15 V to the thin film formed through PEALD. Figure 9a shows the breakdown characteristics according to the nanolaminate thickness. The D(5,5)*12 and D(60,60)*3 specimens showed the lowest breakdown voltage, while the D(10,10)*6 and D(20,20)*3 specimens showed high breakdown voltage characteristics. This characteristic seems to be due to the improvement in the breakdown voltage characteristics of the thin film when the nanolaminate is properly formed. Figure 9b shows the breakdown voltage according to the RTA temperature. Increasing the RTA temperature of the (20,20)*3 HZO nanolaminate specimen from 550 °C to 650 °C reduced the breakdown voltage from 7.15 MV/cm to 4.08 MV/cm. This drop in voltage resulted from the formation of a leakage path formed by excessive crystallization. Depending on the presence or absence of the seed layer in Figure 9c, most of the voltage was maintained until a high value, and the specimen to which the bottom seed layer was applied showed the highest breakdown voltage. Because of the low crystallization temperature of the ZrO_2_ seed layer, crystallization occurs from the bottom, and it is believed that the properties of the thin film are improved.

The P–E hysteresis curve was obtained by applying a voltage of ±5 V as a triangular pulse. The pulse length was 0.0001 s, and the sampling rate was 1 kHz. The shape of the applied pulse and the graphs obtained during the wake-up process over 10^4^ cycles are shown in Figure 10 and Figure 11, respectively. In the wake-up process, cycling was performed by applying a voltage of ±5 V as a rectangular pulse at a frequency of 100 kHz.

An analysis of the P–E hysteresis curve (Figure 11a) shows that the 2 P_r_ value tended to increase with the thickness of the nanolaminate: however, the device with the (60,60)*1 nanolaminate was frequently destroyed during the measurement process, owing to the generation of a relatively large leakage current. This demonstrates the low durability of the device. An analysis of the effect of the heat-treatment temperature (Figure 11b) reveals that the specimen heat treated at 550 °C exhibits the highest 2 P_r_ value. The ferroelectric properties of the specimen heat treated at 650 °C degraded during the crystallization of the HZO thin film with the formation of the M phase, as shown in the XRD patterns in Figure 8 and Figure 9. The 2 P_r_ value of this specimen also tended to decrease. As shown in Figure 11c, specimens without the seed layer (W/O) and those with a top seed layer exhibited similar properties. In addition, the properties of specimens with the bottom seed layer were similar to those with a dual seed layer. This similarity is thought to arise from the difference in the interface formed between the seed layers and the TiN lower electrode. In the W/O and top seed layer specimens, a reaction occurs between the TiN lower electrode and the HfO_2_ layer. Similarly, a reaction between the TiN lower electrode and the ZrO_2_ seed layer occurs in the bottom seed layer and dual seed layer specimens. The ZrO_2_ seed layer at the bottom can be crystallized at a relatively low temperature, and in this case, it acts as a nucleation layer to promote the nucleation of the HZO ferroelectric layer, resulting in a relatively high remnant polarization value.

The built-in field is removed as the oxygen vacancies in the ferroelectric layer are redistributed uniformly during the cycling process, in which pulses are repeatedly applied. A wake-up effect occurs during the cycling process in which the P_r_ value increases; however, fatigue occurs when the device is damaged owing to excessive cycling, while the P_r_ value decreases as the conductive path is formed [30,31].

The reliability of the device was assessed using the fatigue endurance test to evaluate whether the durability and electrical properties of the device are preserved even after multiple cycles. MFM devices were prepared by applying each seed layer to the (20,20)*3 nanolaminate and heat treating at 550 °C. These devices were used to assess the effect of the seed layer on durability. The fatigue endurance test was performed by applying 10^8^ cycles of the rectangular pulse (Figure 9a) before conducting measurements using the triangular pulse (Figure 9b).

Figure 12 shows the results of the fatigue endurance test. The device without the seed layer and that with the top seed layer maintained their initial durability characteristics over 10^7^ cycles. In addition, the devices with the bottom and dual seed layers remained stable without the destruction of the device even after 10^8^ cycles.

Moreover, the device without the seed layer and that with the top seed layer underwent a fatigue phenomenon, in which the remnant polarization decreased without the wake-up effect. By contrast, the wake-up effect was observed in the devices with the bottom and dual seed layers. The fatigue phenomenon occurred upon reaching the maximum P_r_ value after 10^5^ cycles. The lower TiN electrode and the HfO_2_ layer form an interface in the specimens without the seed layer and those with the top seed layer, whereas the specimens with the bottom and dual seed layers form an interface between the bottom TiN electrode and the ZrO_2_ layer. More oxygen vacancies are thought to be formed between the TiN lower electrode and ZrO_2_. The wake-up effect that improves the P_r_ value occurs upon redistribution of these oxygen vacancies during the cycling process.

## 4. Conclusions

This study aimed to improve the ferroelectric properties of HZO ferroelectric materials. HZO ferroelectric thin films were formed by adjusting the HZO nanolaminate thickness and RTA conditions, as well as by applying a seed layer. The properties of the thin films were evaluated by ellipsometry, TEM, XPS, and XRD. In addition, an MFM device was fabricated by forming a TiN upper electrode, and the semiconductor properties of this device were analyzed.

The remnant polarization of the MFM device increased with the HZO nanolaminate thickness, but the durability of the device deteriorated. The breakdown voltage and residual polarization values of the specimens heat treated at 550 °C were higher than those of the specimens heat treated at different temperatures. The (20,20)*3 device presented a residual polarization of 23.94 μC/cm^2^ when heat treated at 550 °C, whereas upon application of the ZrO_2_ seed layer, the D(20,20)*3 device demonstrated an improved residual polarization of 28.18 μC/cm^2^. In addition, the properties of the specimens without a seed layer and those with the top seed layer were similar. Moreover, the properties of the specimens with the bottom and dual seed layers were similar. In the fatigue endurance test, the specimens with the bottom or dual seed layer exhibited the wake-up effect. The device maintained excellent durability even after 10^8^ cycles.

## Figures and Tables

**Figure 1 materials-16-01959-f001:**
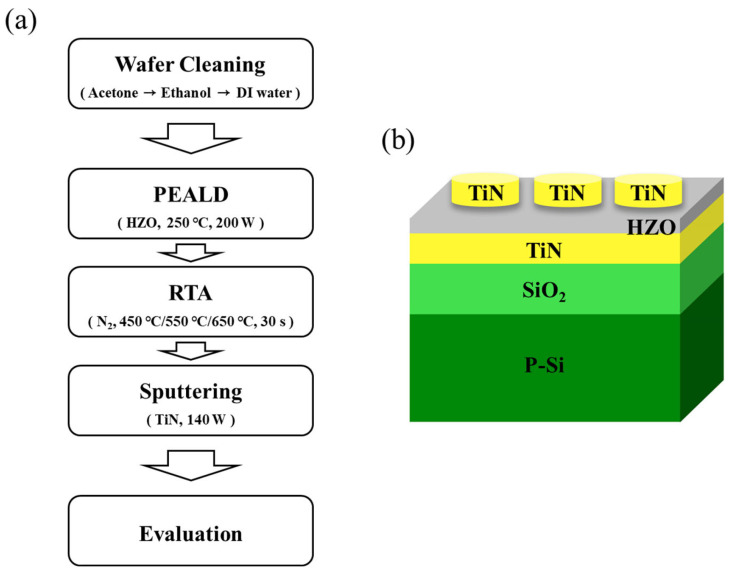
Schematics of (**a**) MFM device production and (**b**) MFM device structure.

**Figure 2 materials-16-01959-f002:**
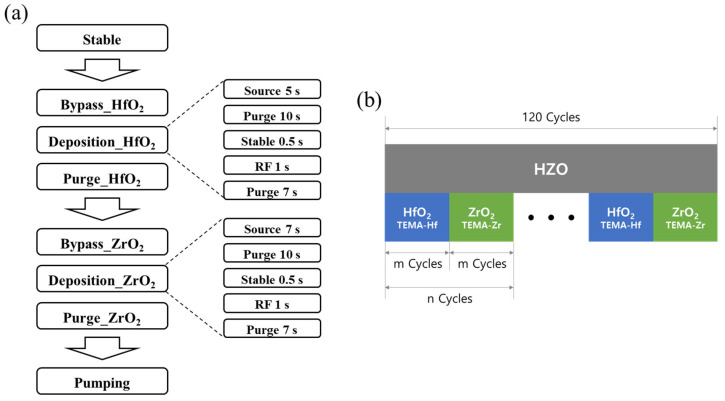
Schematic of (**a**) ALD deposition process (**b**) HZO thin film deposition.

**Figure 3 materials-16-01959-f003:**
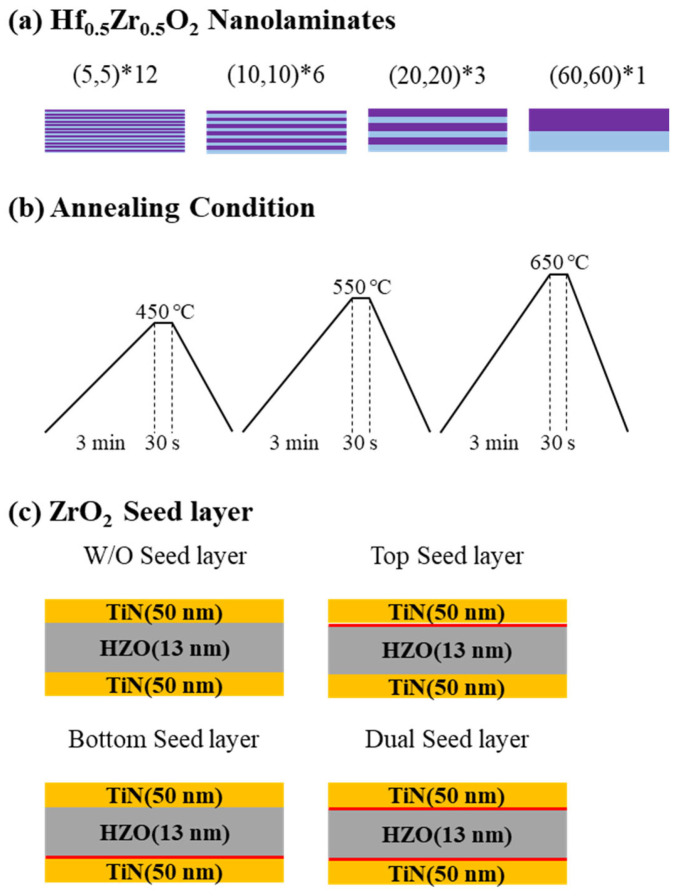
Three fabrication variables: (**a**) Hf_0.5_Zr_0.5_O_2_ nanolaminate, (**b**) annealing condition, and (**c**) ZrO_2_ seed layer.

**Figure 4 materials-16-01959-f004:**
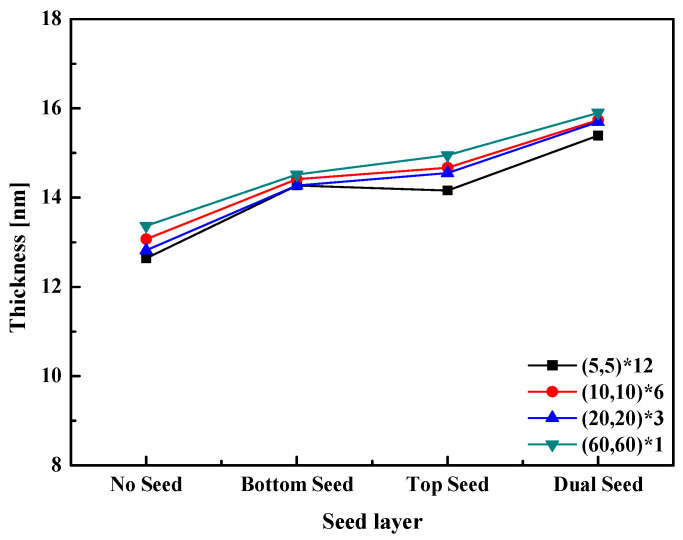
Thickness of the HZO thin films of the HZO nanolaminates.

**Figure 5 materials-16-01959-f005:**
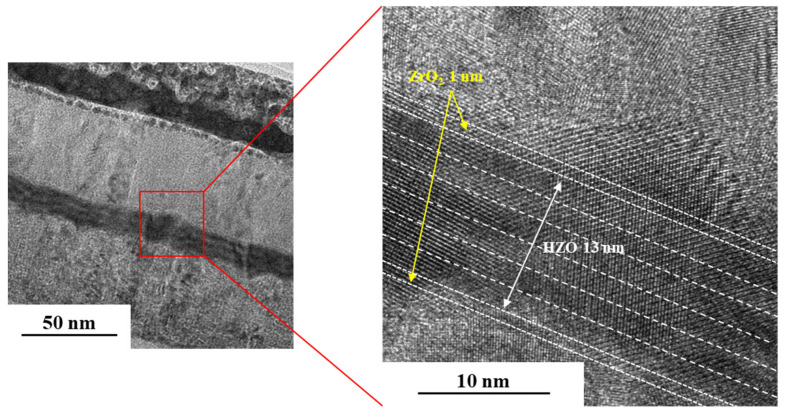
Cross-sectional TEM image of the D(20,20)*3 specimen annealed at 550 °C.

**Figure 6 materials-16-01959-f006:**
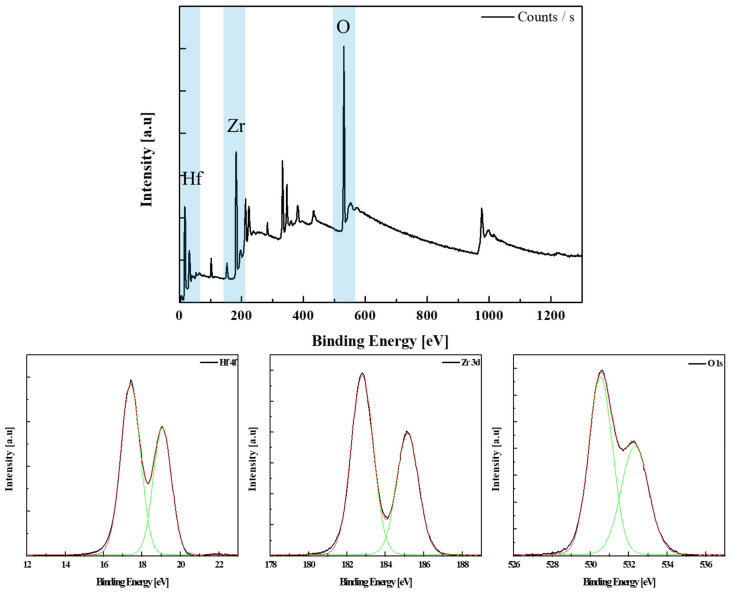
XPS spectra of the (20,20)*3 specimen.

**Figure 7 materials-16-01959-f007:**
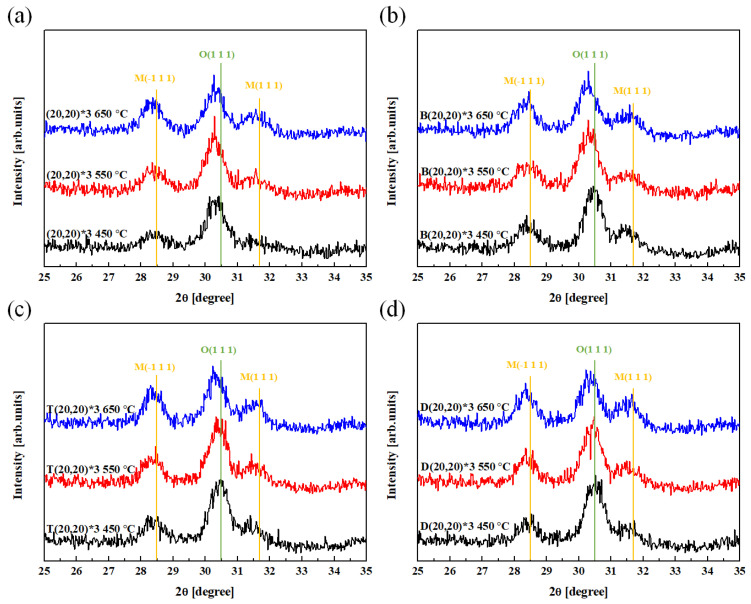
XRD patterns of the (**a**) (20,20)*3, (**b**) B(20,20)*3, (**c**) T(20,20)*3, and (**d**) D(20,20)*3 specimens as a function of the RTA temperature.

**Figure 8 materials-16-01959-f008:**
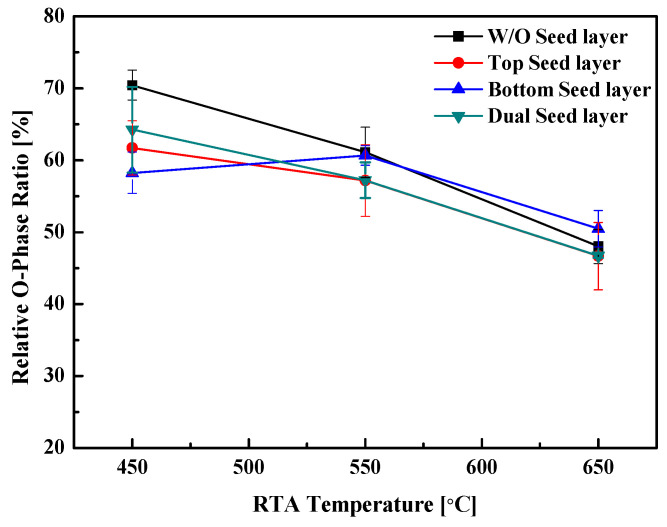
Relative ratio of the O/T/C phase of the four types of specimens according to the application of the seed layer.

**Figure 9 materials-16-01959-f009:**
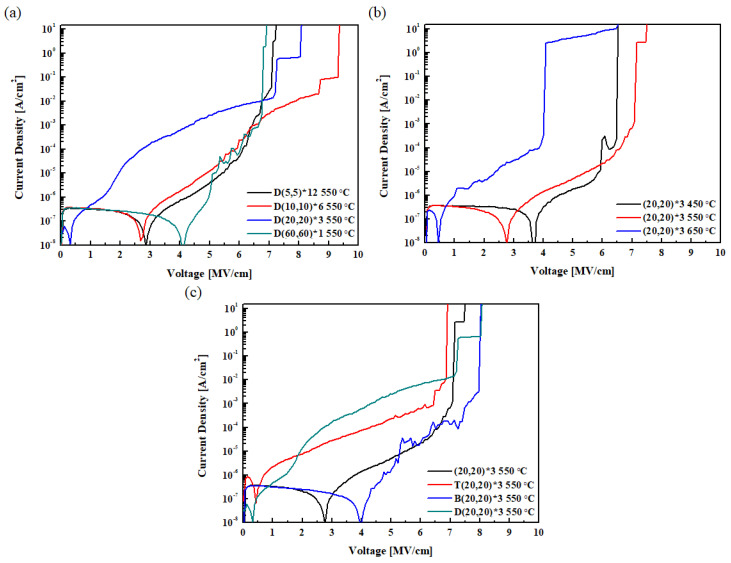
Breakdown voltage according to the variables: (**a**) HZO nanolaminate thickness, (**b**) annealing conditions, and (**c**) seed layer.

**Figure 10 materials-16-01959-f010:**
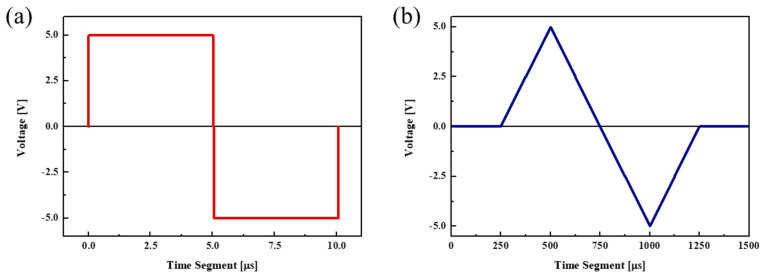
Pulse shapes used in the characteristics analysis: (**a**) rectangular and (**b**) triangular pulses.

**Figure 11 materials-16-01959-f011:**
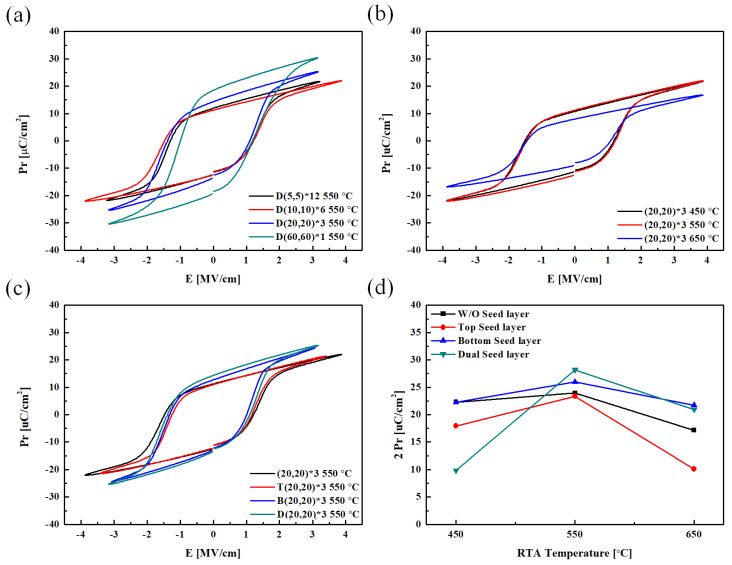
P−E hysteresis curves according to the variables: (**a**) HZO nanolaminate thickness, (**b**) annealing condition, and (**c**) seed layer. (**d**) 2 P_r_ value according to the seed layer and annealing condition.

**Figure 12 materials-16-01959-f012:**
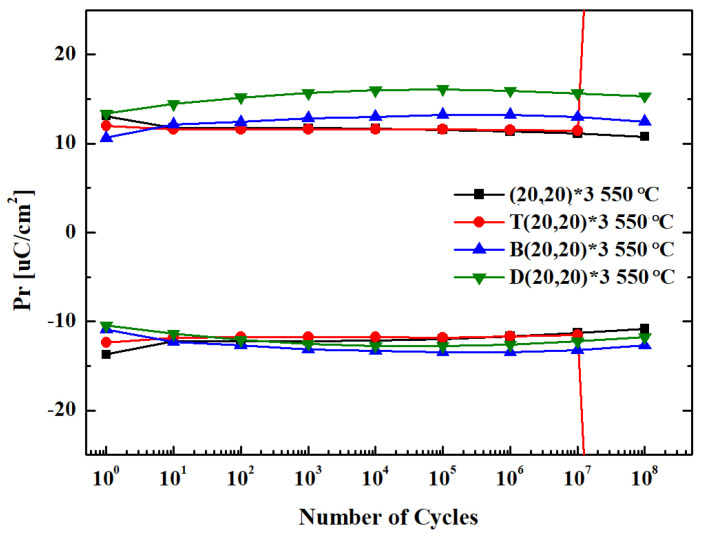
Fatigue endurance of the 550 °C–RTA-treated specimens in the absence and presence of the seed layer.

## Data Availability

Not applicable.

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
