# Peer review of "Effect of a ZrO2 Seed Layer on an Hf0.5Zr0.5O2 Ferroelectric Device Fabricated via Plasma Enhanced Atomic Layer Deposition"

_materials, 2023, doi:10.3390/ma16051959_

Round 1

Reviewer 1 Report

The present paper reported by Ji Na Song. Et.al. Studied the electrical properties of provided ZrO2 Seed Layer on an Hf0.5Zr0.5O2 Ferroelectric. This layer was provided using Plasma Enhanced Atomic Layer Deposition. I am providing some comments that need to be addressed. Overall, this paper can be acceptable for publication in the Journal of Materials after addressing the following point:

1-     Clarify the formation mechanism of ZrO2 Layer on an Hf0.5Zr0.5O2 with more details.

2-     Introduction section should be modified and reflect clear enough background and more explanation about nanomaterials. (Cite Ref: -Journal of Alloys and Compounds 899, 163379, 2022. - Inorganic Chemistry Frontiers 8 (11), 2735-2748, 2021.)

3-     The quality of figures is not suitable.

4-     The surface area and porosity of samples should be investigated using BET-BJH methods.

5-     The reference peaks and (HKL) miller indices should be specified for XRD patterns.

6-     Correlate between XRD result and HR-TEM.

7-     It is better to choose a more attractive title for the paper.

8-     Add AFM analysis for study the surface topography of prepared layers.

9-     Add mapping EDS for better specifying the elemental distribution. 

Author Response

Reviewer #1:

The present paper reported by Ji Na Song. Et.al. Studied the electrical properties of provided ZrO2 Seed Layer on an Hf0.5Zr0.5O2 Ferroelectric. This layer was provided using Plasma Enhanced Atomic Layer Deposition. I am providing some comments that need to be addressed. Overall, this paper can be acceptable for publication in the Journal of Materials after addressing the following point:

  1. Clarify the formation mechanism of ZrO2 Layer on an Hf0.5Zr0.5O2 with more details.

  - To further clarify the formation of the ZrO2 seed layer and HZO ferroelectric layer, the ZrO2 deposition conditions are added to Figure 1.

  1. Introduction section should be modified and reflect clear enough background and more explanation about nanomaterials. (Cite Ref: -Journal of Alloys and Compounds 899, 163379, 2022. - Inorganic Chemistry Frontiers 8 (11), 2735-2748, 2021.)

- A total of 7 references were added to the text, including Journal of Alloys and Compounds 899, 163379, 2022.

  1. The quality of figures is not suitable.

- Pictures with poor visibility were corrected with high-quality pictures to improve visibility, and visibility was improved by increasing the size of pictures. In addition, Figure 9 changed the layout of the figure to make it easier to check.

  1. The surface area and porosity of samples should be investigated using BET-BJH methods.

- sorry. The BET-BJH method, which measures the porosity using a thin film of about 13 nm, seems difficult to measure because it is not available in the surroundings. In the future, additional studies to measure the porosity of thin films should be conducted.

  1. The reference peaks and (HKL) miller indices should be specified for XRD patterns.

- Figure 7 was corrected by adding crystal phases and Miller indices.

  1. Correlate between XRD result and HR-TEM.

- Nanolaminate and crystallinity of the HZO layer were confirmed through the TEM image confirmation in Figure 5, and the crystal phase of each sample was confirmed through HR-XRD analysis. A sentence was added to the text.

  1. It is better to choose a more attractive title for the paper.

- Thank you for your kind comments. Based on your comments, the title has been changed to:

 “Plasma Enhanced Atomic Layer-Deposited ZrO2 Seed Layer on an Hf0.5Zr0.5O2 Ferroelectric Device for an Improved Performance”

  1. Add AFM analysis for study the surface topography of prepared layers.

- Thank you for your kind comments. However, this study focused on the crystallinity and electrical properties of the HZO ferroelectric layer according to the formation method rather than the surface study.

  1. Add mapping EDS for better specifying the elemental distribution. 

- Since it deals with a 13 nm thin film, it is difficult to confirm the elemental distribution using SEM-EDS. Analysis using TEM-EDS is necessary to confirm the accurate element distribution, but equipment that can confirm the element distribution of each layer in atomic units of 1 nm or less is currently difficult to use. Therefore, in this study, the overall element content was analyzed through XPS.

Reviewer 2 Report

In their manuscript “Effect of a ZrO2 Seed Layer on an Hf0.5Zr0.5O2 Ferroelectric Device Fabricated via Plasma Enhanced Atomic Layer Deposition“, the authors investigate many influences on the performance of superlattice-based HZO stacks. While the topic is quite interesting, the here presented data does not provided clear new insights. In addition, some major points that need to be addressed before publishing:

11)      The first three paragraphs of the introduction are void of any citations. Please add sources for major claims. Also add a citation for the discovery of HfO2 in 2011. (The first patent was filed in 2008.)

22)      The authors state that they use a RTA before top electrode deposition to crystallize the film. Why was it decided to anneal it before top electrode deposition, even though it is known from literature that this results in worse ferroelectric performance?

33)      How can the thickness of the superlattices (nanolaminates) differ, if the number of cycles is identical? Is this really a PE-ALD process, or do you have a PE-CVD process? The reaction of the two materials has not been reported elsewhere and they are known to form solid solutions. Also, if interfaces are formed, the smaller sublayers (e.g. 5,5) should result in thicker films.

44)      How do the authors define the sublayers in the TEM image when there is no clear contrast? Also, why do their individual thicknesses differ, if they were deposited via ALD?

55)      Since the XRD patterns are quite noisy, please add error bars to Fig. 8.

66)      In case of breakdown voltage investigation: Why are there major shifts in the minimum position? This normally indicates internal bias fields and the shifts observed here are very large with almost 4V. Furthermore, statistics are completely missing. It can already be seen that the variation in breakdown voltage and shift of D(20,20)*3-550°C varies significantly.

77)      The discussion of ferroelectric properties is unclear to the reader. The changes in PR are very small in all cases except the D(60,60). In addition, all major differences appear to arise from differences in the permittivity. This however is not discussed at all. It is questionable to the reader, if the observed differences are reproducible and what their actual origins are.  

88)      The discussion on built-in fields due to oxygen vacancies is not in context to anything and references are missing.

99)      For the fatigue measurement, what frequency was used? In addition, all conclusions and discussions formed on the presented data appear dubious, as there is almost no difference in behaviour and statistics are missing.

Minor Points:

-          Do not write the explanation of abbreviations in caps (like in the case for RTA)

Author Response

I am pleased to re-submit our revised version of “Plasma Enhanced Atomic Layer-Deposited ZrO2 Seed Layer on an Hf0.5Zr0.5O2 Ferroelectric Device for an Improved Performance” for publication. I appreciated the thorough review and constructive criticisms of the reviewers. I have addressed each of their concerns below and have rewritten sections of the paper to provide clarity. I hope the revision has improved the paper to a level of the reviewers’ satisfaction.

Reviewer #2:

In their manuscript “Effect of a ZrO2 Seed Layer on an Hf0.5Zr0.5O2 Ferroelectric Device Fabricated via Plasma Enhanced Atomic Layer Deposition“, the authors investigate many influences on the performance of superlattice-based HZO stacks. While the topic is quite interesting, the here presented data does not provided clear new insights. In addition, some major points that need to be addressed before publishing:

  1. The first three paragraphs of the introduction are void of any citations. Please add sources for major claims. Also add a citation for the discovery of HfO2 in 2011. (The first patent was filed in 2008.)

- Thank you for the good point. Seven references were added at the beginning of the Introduction.

  1. The authors state that they use a RTA before top electrode deposition to crystallize the film. Why was it decided to anneal it before top electrode deposition, even though it is known from literature that this results in worse ferroelectric performance?

- Thank you for your kind comments. It is known that the PMA method, in which heat treatment is performed after forming the upper electrode, is more advantageous in improving ferroelectric properties. In this study, an experiment was conducted by adopting the PDA method to exclude the capping effect and confirm the ferroelectric properties improved by the seed layer.

  1. How can the thickness of the superlattices (nanolaminates) differ, if the number of cycles is identical? Is this really a PE-ALD process, or do you have a PE-CVD process? The reaction of the two materials has not been reported elsewhere and they are known to form solid solutions. Also, if interfaces are formed, the smaller sublayers (e.g. 5,5) should result in thicker films.

- In this study, deposition was carried out using PEALD, and conditions were established through self-limiting and ALD window experiments before proceeding with the experiment. The conditions are shown in the figure below.

According to the opinions, the content was modified by changing the reaction at the HfO2-ZrO2 Nanolaminate interface to solid solution.

  1. How do the authors define the sublayers in the TEM image when there is no clear contrast? Also, why do their individual thicknesses differ, if they were deposited via ALD?

- After forming a thin film using PEALD, thickness analysis was performed using an ellipsometer, and it was confirmed that the film was formed uniformly. TEM analysis was performed to observe the nanolaminate. Prior to cross-section analysis using TEM, surface treatment is performed using FIB. In this process, a problem arised in that the cross section of the thin film becomes uneven. Observation of nanolaminate is possible through TEM, but in order to secure uniformity, an arbitrary guide line was set in Figure 5 to make it easy to distinguish nanolaminate.

  1. Since the XRD patterns are quite noisy, please add error bars to Fig. 8.

- Thank you for your nice comment. An error bar was added to Figure 8 considering the noise of the XRD pattern.

  1. In case of breakdown voltage investigation: Why are there major shifts in the minimum position? This normally indicates internal bias fields and the shifts observed here are very large with almost 4V. Furthermore, statistics are completely missing. It can already be seen that the variation in breakdown voltage and shift of D(20,20)*3-550°C varies significantly.

- I totally agree with what you said, and it is judged that the current density showing various minimum values is due to the change in the internal field. As a result of repeated measurements several times, similar results were found, and it is thought that additional research is needed in the future on this part.

  1. The discussion of ferroelectric properties is unclear to the reader. The changes in PR are very small in all cases except the D(60,60). In addition, all major differences appear to arise from differences in the permittivity. This however is not discussed at all. It is questionable to the reader, if the observed differences are reproducible and what their actual origins are.  

- The effect of increasing the Pr value according to the presence or absence of the seed layer, which was the ultimate goal of this study, can be confirmed in figure11 (c). Significant results were obtained with about 17.7% improvement in Pr value when the Dual Seed layer was applied compared to when there was no Seed layer. It was judged that this was a result of promoting nucleation due to the low crystallization temperature of ZrO2 when a ZrO2 seed layer existed at the bottom and securing an internal field favorable for obtaining high remanent polarization.

  1. The discussion on built-in fields due to oxygen vacancies is not in context to anything and references are missing.

- In this study, the reason why the fatigue endurance characteristics change according to the seed layer was determined to be the result of the movement of oxygen vacancies. For better understanding, information related to the movement of oxygen vacancies is included in the text. References [29, 30] are included in the description of oxygen vacancies.

  1. For the fatigue measurement, what frequency was used? In addition, all conclusions and discussions formed on the presented data appear dubious, as there is almost no difference in behaviour and statistics are missing.

- The frequency used for fatigue measurement is 100kHz, and the voltage and waveform used can be seen in Figure 10. As mentioned in the text, in the case of specimens with a seed layer at the bottom, a wake-up effect appeared due to the rearrangement of oxygen vacancies, and the characteristics were maintained up to 108 cycles. On the other hand, the specimen without the lower seed layer had relatively few oxygen vacancies, so the wake-up effect did not occur, and the device was destroyed in 107 cycles.

Reviewer 3 Report

1. The electrical properties of TiN film should be added, because it will affect the conductivitive properties of the fabricated devices.

2. How can you control the thicknesses and volumes of HfO2 and ZrO2 in eack cycle? Because you said your films were Hf0.5Zr0.5O2 nanolaminate.

3. What are the B(20,20)*3 and T(20,20)*3? What the differences between the four Figures in Figure 7?

4. Some I-V curves in Figure 9 are poor, authors should explain.

Author Response

Dear Editor:

I am pleased to re-submit our revised version of “Plasma Enhanced Atomic Layer-Deposited ZrO2 Seed Layer on an Hf0.5Zr0.5O2 Ferroelectric Device for an Improved Performance” for publication. I appreciated the thorough review and constructive criticisms of the reviewers. I have addressed each of their concerns below and have rewritten sections of the paper to provide clarity. I hope the revision has improved the paper to a level of the reviewers’ satisfaction.

Reviewers' comments:

Reviewer #3:

  1. The electrical properties of TiN film should be added, because it will affect the conductivitive properties of the fabricated devices.

- The relevant information was added to the end of 2.1 HZO Ferroelectric Device Fabrication of 2. Materials and Methods. The added contents are as follows. 'The resistance of the TiN upper electrode deposited by sputtering is about 29 ohm/sq.'

  1. How can you control the thicknesses and volumes of HfO2 and ZrO2 in eack cycle? Because you said your films were Hf0.5Zr0.5O2 nanolaminate.

- In PEALD, the thickness of the thin film is determined by the number of deposition cycles. With the PEALD used in this study, it was confirmed that 1.18 Å of HfO2 and 1.15 Å of ZrO2 were deposited per cycle. By alternately depositing HfO2 and ZrO2 at a constant cycle rate using PEALD, a Hf0.5Zr0.5O2 thin film consisting of a total of 120 cycles was formed. For related information, details can be found in 2.2. variables of 2. Materials and Methods.

  1. What are the B(20,20)*3 and T(20,20)*3? What the differences between the four Figures in Figure 7?

- In Figure 7, (a) is a specimen without a seed layer, (b), (c), and (d) are a specimen with a seed layer only on the bottom, a specimen with only the top, and a specimen with both the top and bottom, respectively. The result of Figure 8 was derived by calculating the relative O phase ratio through Figure 7. To clarify abbreviations, added content to the last sentence of 2.2. variables in 2. Materials and Methods. The contents are as follows. ‘When the ZrO2 seed layer is on the bottom, it is marked as B, when it is only on the top, it is marked as T, and when it is on both the top and bottom, it is marked as D.’

  1. Some I-V curves in Figure 9 are poor, authors should explain.

- Edited content on page 9. The modified contents are as follows.

‘Figure 9 shows the breakdown voltages measured by applying a DC voltage in in-crements of 0.5 V from 0 V to 15 V to the thin film formed through PEALD. Figure 9 (a) shows breakdown characteristics according to nanolaminate. D(5,5)*12 and D(60,60)*3 specimens showed the lowest breakdown voltage, and in the case of D(10,10)*6 and D(20,20)*3 specimens, voltage characteristics were observed. This characteristic seems to be due to the improvement of the withstand voltage characteristics of the thin film when the nanolaminate is properly formed. Figure 9 (b) shows breakdown voltage according to RTA temperature. Increasing the RTA temperature of the (20,20)*3 HZO nanolaminate specimen from 550 °C to 650 °C reduced the breakdown voltage from 7.15 MV/cm to 4.08 MV/cm. This drop in the voltage resulted from the formation of a leakage path formed by excessive crystallization. In the results according to the presence or absence of the seed layer in Figure 9 (c), most of the voltage was maintained until high, and the specimen to which the bottom seed layer was applied showed the highest breakdown voltage. Due to the low crystallization temperature of the ZrO2 seed layer, crystallization occurs from the bottom, and it is believed that the properties of the thin film are improved.’

In addition, some modifications were made to the arrangement to improve the quality of the Figure.

Reviewer 4 Report

The paper aims to investigate the effect of HZO nanolaminate thickness, the thermal treatment conditions and the seed layer (top-only, bottom-only, dual, no seed layer) to the improvement of the ferroelectric properties of HZO materials. It is in a well written form, which leads to a comprehensive reading. The literature cited is relevant and updated.

Some minor comments to improve the manuscript.

1.     Figures 1, 2, 3, 4, 5, 6, 10, 12 are not mentioned in the text. All figures presented should be mentioned in the text and briefly explained.

2.     At some points, text need rephrasing: for example, in line 85 the “cleaned sequentially untrasonically” should be better read as “ultrasonic-cleaned sequentially”, in line 208 the “frequency was measured” should be better as “the sampling rate”, in Figure 4 X-axis instead of “Seed X” should be better “No Seed” and in Figure 7 it would be better if the “M” and “O” phases are marked in the figure.  

Author Response

Dear Editor:

I am pleased to re-submit our revised version of “Plasma Enhanced Atomic Layer-Deposited ZrO2 Seed Layer on an Hf0.5Zr0.5O2 Ferroelectric Device for an Improved Performance” for publication. I appreciated the thorough review and constructive criticisms of the reviewers. I have addressed each of their concerns below and have rewritten sections of the paper to provide clarity. I hope the revision has improved the paper to a level of the reviewers’ satisfaction.

Reviewer #4:

The paper aims to investigate the effect of HZO nanolaminate thickness, the thermal treatment conditions and the seed layer (top-only, bottom-only, dual, no seed layer) to the improvement of the ferroelectric properties of HZO materials. It is in a well written form, which leads to a comprehensive reading. The literature cited is relevant and updated.

Some minor comments to improve the manuscript.

  1. Figures 1, 2, 3, 4, 5, 6, 10, 12 are not mentioned in the text. All figures presented should be mentioned in the text and briefly explained.

- Thank you for your good point. I have added and edited the relevant part to the text.

  1. At some points, text need rephrasing: for example, in line 85 the “cleaned sequentially untrasonically” should be better read as “ultrasonic-cleaned sequentially”, in line 208 the “frequency was measured” should be better as “the sampling rate”, in Figure 4 X-axis instead of “Seed X” should be better “No Seed” and in Figure 7 it would be better if the “M” and “O” phases are marked in the figure.  

- The content of the text has been revised to reflect what you said, and the picture has been changed.

Round 2

Reviewer 2 Report

While the authors did minor changes to their manuscript, the major issues remain:

1) In their response to former point 3, the now claim that the thickness difference is due to forming a solid solution. However, this does not relate to a change in the thickness of the film. They do not add any evidence to support their claim. Please add an error discussion and reference of relevant literature to support your claim. Other groups have not reported on such a behavior. Nevertheless, I acknowledge the additional provided data from the authors which confirms a PEALD process window.

2) In their response to former point 4, the authors state that the drawn lines in the TEM are arbitrary. Consequently, the TEM image does not reveal a stacked structure; especially since the interfaces are clearly not as straight as suggested by the lines.

3) In their response to former point 6, they state, that these results are repeatable, however, like mentioned in point 6 before, the here presented variance of one sample is already larger than deduced trends by the authors. Additional statistics need to be presented to have a meaningful discussion on this regard.

4) In their response to former point 7, the authors state that fig. 11c provides evidence for the difference in PR depending on the seed layer. However, it can be seen in this figure that the change is dominated by a difference in the slope at high voltages. Consequently there is a difference in the permittivity and the results cannot be compared just by looking at the PR value. THe authors should provide PUND measurements to remove permittivity influences and demonstrate actual differences. In addition, the changes in PR between different samples are quite small and the hysteresis is for some reason not closed. Please provide statistics for a meaningful discussion.

5) In the authors response to former point 9, the frequency of the measurement is stated, but no change to the manuscript was found. In addition, this frequency is quite high, and RC delays are depending on their device size expected. Please add supporting information that provides evidence that the applied voltage drops across the device. Moreover, statistics are still missing and the explanation by the authors is not supported by evidence nor references.

Author Response

Dear Reviewer:

I am pleased to re-submit our revised version of “Plasma Enhanced Atomic Layer-Deposited ZrO2 Seed Layer on an Hf0.5Zr0.5O2 Ferroelectric Device for an Improved Performance” for publication. I appreciated the thorough review and constructive criticisms of the reviewers. I have addressed each of their concerns below and have rewritten sections of the paper to provide clarity. I hope the revision has improved the paper to a level of the reviewers’ satisfaction.

Response 1 :

Thank you very much for the good point. We also agonized over the cause of occurrence in various ways, and in the paper attached below, we confirmed that the thickness of our nanolaminate showed a similar aspect to ours. In the additional paper, high GPC was shown in the case of single oxide, and the smaller the nanolaminate, the lower the GPC. The study determined that this was due to reduced nucleation during the deposition of the alternating layers. It seems that further research is needed to understand this part accurately. Reference 29 was added and inserted.

Response 2 :

Thank you very much for your wonderful and insightful point. As you said, it is true that there is no clearly defined line because a solid solution is formed at the interface. In addition, in order to observe the sample, it was confirmed that the HZO nanolaminate was curved in the process of processing the specimen using the FIB for the cross section. As can be seen on the left side of Figure 5, although there is no guideline, the light- and dark-colored layered structure can be visually checked, so the reader's understanding is provided with a marker line.

Response 3

 As the reviewer noted, it is necessary to produce a large number of samples for meaningful statistics. However, in the current situation, PEALD is reserved and used by other research institutes, so it costs a lot of money. Therefore, please understand that sample production is difficult. 

Response 4 :

Thank you for the good point that if you add the PUND measurement, it is possible to measure the dielectric constant effect is removed. However, we currently do not have measurable samples. As in the previous question 3, it is difficult to drive PE-ALD. If this part is written in a follow-up thesis, I will deal with it together. Thank you for your kind comments and consideration.

Response 5 :

 The electrical characteristics measurement conditions you mentioned are written in lines 225-229 of the text. The specifications of the equipment used in this study were configured by adding a PMU Unit to the Keithley 4200A-SCS equipment. The specifications of the equipment used are attached below. Since the size of the Dot of the sample is 400um, 100kHz, 5V, the equipment used is a facility capable of driving +-40V and 800mA in ns, so it is thought that RC delay will not occur.

Reviewer 3 Report

This paper can be accepted as it is. 

Author Response

As you said, some of the contents of the thesis, such as spelling, were corrected and re-edited.

An English proofread from an accredited institution must be attached.
